

# Inferring particle dynamics in the Mediterranean Sea from in-situ pump POC and chloropigment data using Bayesian statistics

Weilei Wang[1], Cindy Lee[2], and François Primeau[1]

[1]Department of Earth System Science, University of California at Irvine, Irvine, CA 92617, USA
[2]School of Marine and Atmospheric Sciences, Stony Brook University, Stony Brook, NY 11790-5000, USA

**Correspondence:** Weilei Wang (wang.weilei@uci.edu)

**Abstract.** Chloropigment and particulate organic carbon (POC) concentration data collected using in-situ large-volume pumps during the MedFlux project in the Mediterranean Sea in May 2005 provided an opportunity to estimate rate constants that control the fate of particles and specifically chloropigments in the water column. Additionally, comparisons to thorium and chloropigment data from settling-velocity (SV) sediment traps at the same site enabled us to distinguish between the influence of the sampling method used vs. the tracer used on particle dynamic rate constants. Here we introduce a Bayesian statistical inversion method that combines the data with a new box model and has the capacity to infer rate constants for POC respiration/dissolution, chlorophyll and phaeopigment degradation, and particle aggregation and disaggregation. The estimated small-particle (1-70 $\mu$m) POC respiration rate constant was $1.25^{+0.55}_{-0.38}$ yr$^{-1}$ (0.80 yr). For this data set, the rate constants for chlorophyll (Chl) degradation to phaeopigments and phaeopigment respiration were not well constrained. The estimated aggregation and disaggregation rate constants were $7.65^{+3.35}_{-2.33}$ (0.13 yr) and $106.09^{+39.13}_{-28.59}$ yr$^{-1}$ (0.01 yr), respectively, which indicates that particle aggregation and disaggregation were extensive at the studied depths (125-750 m) in May after the spring bloom had ended and flux was low.

## 1 Introduction

Sinking particles transport carbon to the ocean interior at a globally significant rate of 4-13 Pg C (Lima et al., 2014), one of the few natural processes removing CO$_2$ from the atmosphere for periods significant to climate change. However, most organic matter produced by photosynthesis in the euphotic zone, a key source of marine particles, is consumed or respired in the upper ocean. Only about 10% of the total organic matter produced is transported into the deep ocean and/or sediment by sinking particles (Lee and Wakeham, 1988). Theoretically, Particle density and size determine particle sinking speed according to Stoke's law, and thus residence time in the water column (McCave, 1975; Clegg and Whitfield, 1991; Armstrong et al., 2002). However, processes such as aggregation and disaggregation can alter this relationship and ultimately influence particle transfer efficiency (e.g. McDonnell and Buesseler (2010).

Historically, the quantitative dynamics of particle aggregation and disaggregation processes has been studied using thorium isotopes (Bacon and Anderson, 1982; Clegg and Whitfield, 1991; Clegg et al., 1991; Murnane, 1994; Murnane et al., 1996; Cochran et al., 1993; Cochran et al., 2000), and much of what we currently know about the dynamics of these processes is from

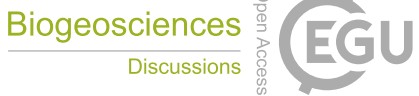



work with particulate thorium. Thorium radioisotopes have unique advantages, e.g., multiple radionuclides can share the same chemical characteristics but have different decay half-lives, which offers extra degrees of freedom. In addition, environmental sources of radionuclides are well-documented. However, thorium tracers also suffer several inherent, disadvantages, e.g., 1) different particle components have different degrees of affinity, to thorium (Roy-Barman et al., 2005); 2) low particulate

thorium activity, especially in the oligotrophic deep ocean, limits their application; and 3) dissolved thorium activity is orders of magnitude higher than particulate activity, thus in models, dissolved thorium influences the adsorption-desorption balance more than particulate thorium does (Wang et al., 2016). New tracers that can overcome those disadvantages should be explored to get a more complete picture of particle dynamics in the ocean.

Chloropigment (Chl *a* and its degradation products: pheophorbide, pyropheophorbide, and pheophytin) tracers were first

proposed as an alternative for thorium isotopes in tracing particle dynamics by Wang et al. (2017), who used chloropigments sampled using Indented Rotating Sphere (IRS) sinking velocity (SV) traps to study particle aggregation and disaggregation. In that study, particle sinking speeds were measured using SV traps, and sinking particles were grouped into two categories based on their sinking speed. A "two-layer" model was applied to estimate particle aggregation and disaggregation, and particulate organic matter respiration rate constants. Compared to thorium as a tracer, one distinct advantage of chloropigments is that

due to their insolubility in water, they do not adsorb to or desorb from particles. Chloropigments are also an integral part of most marine particles formed in the euphotic zone. Wang et al. (2017) compared rate constants obtained using chloropigment concentrations to their counterparts obtained using thorium isotope activities; chloropigments and thorium had been measured by the same trap. There were clear differences: aggregation and disaggregation rate constants estimated from chloropigments were orders of magnitude higher than those from thorium tracers. These results were thought to be due to the differences in the

way thorium and chlorophyll exist on and within particles.

SV sediment traps have several advantages compared to other techniques, such as time-series sediment traps or large volume pumps, because SV sediment traps can measure both sinking particle flux and particle sinking velocity (Peterson et al., 2005, 2009; Armstrong et al., 2009). By using SV trap data, we can avoid any assumptions that link particle size to particle sinking velocity, assumptions that may not be true in the ocean (e.g., McDonnell and Buesseler, 2010). However, due to the lack of

easy commercial availability, SV sediment traps have not been widely used in oceanographic surveys. In addition, interpretation of aggregation and disaggregation derived from SV trap seems awkward, since traditionally it is thought that aggregation is the process of small particles colliding with other particles and becoming larger, and disaggregation is the process of larger particles breaking up into smaller particles. Whereas, particles in fast sinking categories are a combination of large and small particles, because some small particles have high sinking speed, and vice versa (McDonnell and Buesseler, 2010).

In-situ pumps can collect both large and small particles by filtering the seawater through filters or meshes of different size (Bishop et al., 2012). In terms of flux measurements, these pumps have a drawback in that they take a snapshot of the water column only over a 1-2 day period and the different depths sampled are not always sampled on the same day due to logistical constraints. In addition, particles can break up on filters and may not reflect their in-situ size. However, they are much cheaper





than traps, have simpler electronics, and are easier to deploy and recover. Unfortunately, there has been no generally accepted method to convert pump-derived concentrations to flux.

As mentioned above, chloropigments measured in sediment trap samples have been used in a previous study as tracers to study particle dynamics (Wang et al., 2017). Knowing both sinking flux and sinking velocity, a conceptual model was relatively

easy to build. However, *in situ* particle sinking velocity is rarely measured, thus, different techniques are required to model pigment concentrations sampled using large-volume pumps at multiple depths. In the following, chloropigments sampled using large-volume pumps were used to study particle aggregation, disaggregation, and particulate organic carbon respiration rate constants. The objectives of this new work were to 1) introduce a new method that allows the use of pump-sampled pigment tracers (or other geochemical tracers) to study particle dynamics, and 2) compare particle exchange rate constants with two

other studies at the same site. The other two studies used thorium (Wang et al., 2016) and chloropigment (Wang et al., 2017) tracers collected using the same sampling technique. This study and (Wang et al., 2017) used the same chloropigment tracer but sampled using different techniques, i.e., SV trap versus large volume pumps. These inter-comparisons enabled us to investigate whether sampling techniques or the tracer itself imposed a stronger constraint on modeled rate constants.

## 2 Methods

### 15 2.1 Sampling Site

Samples were collected using in-situ pumps as part of the MedFlux project at the French JGOFS DYFAMED (Dynamics of Atmospheric Fluxes in the Mediterranean) site (4320'N, 740'E) in the Mediterranean Sea in May, 2005 (Cochran et al., 2009). Lee et al. (2009) describe the DYFAMED site and why it was chosen for the MedFlux study. This site in the Ligurian Sea has an average water depth of 2300 m and is 53 km from the coast at Nice. It is traditionally treated as an open ocean site because

the alongshore Ligurian current cuts off most terrestrial influence (Marty et al., 2002). This site is characterized by strong winter mixing and low winter primary production, followed by a strong phytoplankton spring bloom from March to April. The spring bloom is terminated by summer stratification and thus a shorter supply of nutrients. A smaller phytoplankton bloom in fall is promoted by the decrease in stratification at that time.

### 2.2 Sampling and analyses

Medflux chloropigment and POC sampling methods, analytical procedures, and pump concentration data have been described and discussed previously in Abramson et al. (2010). Essentially, large volume in-situ pumps (Challenger Oceanic) were used to collect particle samples that were separated into two groups based on their size. Seawater flowed first through a 70-$\mu$m Nitex screen that retained larger (>70 $\mu$m) particles, and then through a 1-$\mu$m quartz microfiber filter that retained smaller (1-70 $\mu$m) particles. Once retrieved, half of each 142-mm filter was frozen for later chloropigment analysis. Material collected on the

70-$\mu$m filter is typically thought to be more representative of sinking particulate matter, whereas material passing through the larger filter and collected on the 1-$\mu$m filter is typically thought to be a more slowly settling fraction (Cochran and Masqué,





2003). As discussed in Abramson et al. (2010), if Stokes' law were applied to these particles, the 70-$\mu$m cutoff would be roughly equivalent to a division between particles settling at speeds less (slowly sinking) and more (faster sinking) than 0.5 m d$^{-1}$.

## 2.3 Data selection

Medflux samples in the euphotic zone (depth <80 m) and below the nepheloid layer (depth > 2000 m) were excluded to reflect the export zone and to eliminate the influence of primary production and sediment resuspension. The original data used in our model are shown in Table 1. The MedFlux site was surveyed during multiple cruises in 2003 and 2005, however, only data from pump samples collected in late May 2005 are used in this study, because >70-$\mu$m-size particles were only collected at that time.

## 10  2.4 Conceptual models

We modified the conceptual model used in Wang et al. (2017) to tailor it for pump data as shown in Fig. 1. Specifically, we changed the slow- and fast-sinking particle groups to smaller and larger particle groups. Particle sinking speeds ($\omega$) were no longer available, therefore, we assumed that small particles do not sink appreciably as has been commonly assumed in similar models (Murnane et al., 1996; Marchal and Lam, 2012). The transport of tracers downward into the water column by

sinking particles is modeled using a transport operator, $\mathbf{S}$, which can be written in matrix form by discretizing the sinking flux divergence, $\frac{\partial(\omega\mathbf{c})}{\partial z}$ , using finite differences. If we discrete the water column into 7 layers, the concentration of a tracer can be organized into a vector, $\mathbf{c}$, and the discretized flux divergence is given by $\mathbf{Sc}$ where $\mathbf{S}$ is an 7x7 matrix.

The interpretation of the conceptual model is similar to the previous version. For example, small and large particles exchange particle components via aggregation and disaggregation: larger particles disaggregate to form smaller particles, and smaller

particles aggregate to become larger particles. Chl that originates in small phytoplankton is incorporated into larger particles via aggregation, and phaeopigments that are products of zooplankton digestion can be associated with smaller particles via particle disaggregation (Abramson et al., 2010). In addition, some phaeopigments are products of microbial Chl degradation, therefore, they can be produced directly in small particles ($d_1$). Chl also can degrade to colorless components, disappearing from our analytical window, but we ignore this process as was done in Wang et al. (2017). Small-sized POC and phaeopigments

can be respired to $CO_2$ ($d_2$ and $d_3$, respectively).

## 2.5 Mathematical descriptions

We assume first-order reaction kinetics for aggregation and disaggregation, which is a gross simplification since the real reaction kinetics is not known. Our objective here, however, is to compare how different sampling methods (SV sediment trap versus large volume pump) can influence inferred particle dynamics when the same tracer is used, and to compare how

different tracers (thorium versus pigments) can influence inferred particle dynamics when the same sampling method is used.



We thus use the same mathematical method and conceptual model found in other published studies (see Murnane et al. (1990); Marchal and Lam (2012); Cochran et al. (1993); Wang et al. (2016, 2017)).

For large size POC, we have the following governing equation:

$$\frac{d[\mathbf{POC_l}]}{dt} = \underbrace{\beta_1[\mathbf{POC_s}]}_{Aggregation} - \underbrace{\beta_{-1}[\mathbf{POC_l}]}_{Disaggregation} - \underbrace{\mathbf{S}[\mathbf{POC_l}]}_{Sinking}, \tag{1}$$

and for small size POC, we have the following governing equation:

$$\frac{d[\mathbf{POC_s}]}{dt} = \underbrace{\beta_{-1}[\mathbf{POC_l}]}_{Disaggregation} - \underbrace{(\beta_1 + d_2)[\mathbf{POC_s}]}_{Aggregation(\beta_1)\ and\ Respiration(d_2)}, \tag{2}$$

where the subscripts $l$ and $s$ denote large and small particulate POC, respectively. The bold variables such as $[\mathbf{POC_x}]$ and $[\mathbf{Chl_x}]$ (appearing in subsequent equations) represent $7 \times 1$ vectors, for the concentrations at each discrete depth level. $\beta_1$ and $\beta_{-1}$ are aggregation and disaggregation rate constants, respectively. We neglect advection and diffusive transport of particles

because their influence is assumed to be small compared to other source and sink terms (Savoye et al., 2006).

Chl, an essential pigment that is produced during phytoplankton photosynthesis, is assumed to originate only in smaller particles. This is a reasonable assumption considering the sampling time (May 2005). At that time, the spring bloom being over, the diatom density should be low so that primary production should be dominated by small phytoplankton. Table 1 also shows that Chl-*a* concentrations were very low at the sampling time, indicating a low primary production condition. We then

assume that any Chl found in large particles is from small particle aggregation. We also assume Chl has a two-step degradation: it first degrades to pheopigments ($d_1$), which will then further degrade to $CO_2$ ($d_3$). The governing equations for pigments are as follow.

$$\frac{d[\mathbf{Chl_l}]}{dt} = \underbrace{\beta_1[\mathbf{Chl_s}]}_{Aggregation} - \underbrace{\beta_{-1}[\mathbf{Chl_l}]}_{Disaggregation} - \underbrace{\mathbf{S}[\mathbf{Chl_l}]}_{Sinking}, \tag{3}$$

$$\frac{d[\mathbf{Chl_s}]}{dt} = \underbrace{\beta_{-1}[\mathbf{Chl_l}]}_{disaggregation} - \underbrace{(\beta_1 + d_1)[\mathbf{Chl_s}]}_{Aggregation(\beta_1)\ and\ Degradation(d_1)}, \tag{4}$$

$$\frac{d[\mathbf{Phy_l}]}{dt} = \underbrace{\beta_1[\mathbf{Phy_s}]}_{Aggregation} - \underbrace{\beta_{-1}[\mathbf{Phy_l}]}_{Disaggregation} - \underbrace{\mathbf{S}[\mathbf{Phy_l}]}_{Sinking}, \tag{5}$$

$$\frac{d[\mathbf{Phy_s}]}{dt} = \underbrace{\beta_{-1}[\mathbf{Phy_l}]}_{Disaggregation} + \underbrace{d_1[\mathbf{Chl_s}]}_{Degradation} - \underbrace{(\beta_1 + d_3)[\mathbf{Phy_s}]}_{Aggregation(\beta_1)\ and\ Degradation(d_3)}. \tag{6}$$

Here we assume the system is at steady state, just as has been done in Wang et al. (2016, 2017) and almost all similar work, e.g.(Murnane et al., 1990; Clegg and Whitfield, 1991; Marchal and Lam, 2012; Lerner et al., 2016, 2017), to make the results





comparable. At steady state, the time derivative terms $\frac{d()}{dt}$ on the left hand of Eq.1 through Eq. 6 vanish and the resulting discretized equations can be written as a system of 42 equations in 42 unknowns (6 tracers in 7 discrete layers). In matrix form we have

$$\mathbf{Jc} = 0, \tag{7}$$

where ($\mathbf{c^T} = [\mathbf{POC_l}, \mathbf{POC_s}, \mathbf{Chl_l}, \mathbf{Chl_s}, \mathbf{Phy_l}, \mathbf{Phy_s}]$) is a $1 \times 42$ row vector and $\mathbf{J}$ is a sparse $42 \times 42$ matrix. To obtain non-trivial solutions for Eq.7, we prescribe the concentration of each tracer in the top layer. This can be achieved by partitioning $\mathbf{c}$ into a part corresponding to the concentrations in the upper most box $\mathbf{c_u}$, and a part corresponding to the remaining 7 boxes, $\mathbf{c_l}$ . Applying the same partitioning to $\mathbf{J}$ we get

$$\mathbf{J} = \begin{bmatrix} \mathbf{J_{lu}} & \mathbf{J_{ll}} \\ \mathbf{J_{uu}} & \mathbf{J_{ul}} \end{bmatrix}. \tag{8}$$

The governing system of equations for the $N = 42$ remaining unknowns is

$$\mathbf{J_{ll}c_l} + \mathbf{J_{lu}c_u} = 0. \tag{9}$$

The solution to Eq.9 can be obtained by direct matrix inversion:

$$\mathbf{c_l} = \mathbf{J}_{ll}^{-1}(\mathbf{J_{lu}c_{u0}}), \tag{10}$$

where $\mathbf{c_{u0}}$ is the upper-most concentration measurements.

The matrix $\mathbf{J}$ is a function of $k = 5$ unknown parameters, $[\beta, \beta_{-1}, d_1, d_2, d_3]$, which makes the solution $\mathbf{c_l}$ an implicit function of these parameters. We estimate these parameters using a two-level Bayesian process. At the first level we estimate the vector of parameters $\mathbf{p} = \log[\beta, \beta_{-1}, d_1, d_2, d_3]$ by assigning a normal probability distribution to the prior probability for $\mathbf{p}$ as well as to the deviations of $\mathbf{c_l}(\mathbf{p})$ from the observations $\mathbf{c_{l0}}$. The resulting logarithm of the posterior is given by $-0.5f(\mathbf{p}|\Gamma, \Lambda) + constant$, with

$$f(\mathbf{p}|\Gamma, \Lambda) = \Gamma(\mathbf{c_l}(\mathbf{p}) - \mathbf{c_{l0}})'\mathbf{\Sigma_d^{-1}}(\mathbf{c_l}(\mathbf{p}) - \mathbf{c_{l0}}) + \Lambda(\mathbf{p} - \mathbf{p_0})'\mathbf{\Sigma_p^{-1}}(\mathbf{p} - \mathbf{p_0}), \tag{11}$$

where $\mathbf{\Sigma_d}/\Gamma$ and $\mathbf{\Sigma_p}/\Lambda$ are the covariance matrices for the likelihood and the prior. The hyper-parameters, $\Gamma$ and $\Lambda$ scale the data and prior precisions. The relative size of $\Gamma$ and $\Lambda$ controls the relative importance of the prior and likelihood function in determining the location of the maximum of the posterior probability for the parameters. Their overall magnitude influences the size of the posterior errorbars. At the first level, we find the most probable parameter values (conditioned on assumed values

for $\Gamma$ and $\Lambda$) by minimizing $f$ using a trust-region algorithm as implemented in the Matlab's fminunc function. By optimizing the logarithm of the parameters rather than the parameters themselves we avoid the possibility that the optimization routine will propose meaningless negative rate constants. After conditioning on the model the errors are expected to be independent so that




$\Sigma_{\mathbf{d}}$ is a diagonal matrix, whose diagonal elements are the squares of the measurement standard deviations of corresponding data. The hyper parameter, $\Gamma$, rescales these variances to account for the fact that model errors can also contribute to the misfits. We also assign a diagonal matrix to $\Sigma_{\mathbf{p}}$, which is equivalent to assuming that a priori knowledge of one parameter does not provide any information on the value of another parameter. We assigned large variance for each parameter (Table 2) to allow them change in wide ranges. Ultimately, only the relative size of the prior parameter precisions matter because we rescale the prior covariance matrix with the hyperv paramter, $\Lambda$. The trust-region optimization algorithm is very efficient because we are able to provide the gradient and hessian matrix for $f(\mathbf{p}|\Gamma,\Lambda)$. Typically, we obtain the optimal parameters in less than 50 iterations.

At the second level we estimate $\Gamma$ and $\Lambda$ by maximizing Bayesian evidence (MacKay, 1992),

$$Z(\Gamma,\Lambda) = \int d\mathbf{p}\, \mathrm{prob}(\mathbf{p}|\mathbf{c_{l0}}), \tag{12}$$

which is equivalent to maximizing the likelihood for $\Gamma$ and $\Lambda$. A good approximation to the logarithm of the evidence is given by

$$\log(Z(\Gamma,\Lambda)) = -f(\hat{\mathbf{p}}) - \frac{1}{2}\log(\det(\mathbf{A})) + \frac{k}{2}\log(\Lambda) + \frac{N}{2}\log(\Gamma) + const. \tag{13}$$

where $\hat{\mathbf{p}}$ is the vector of optimal parameter values estimated at the first level for a prescribed $\Gamma$ and $\Lambda$. $\mathbf{A}$ is the $k \times k$ Hessian matrix computed by taking the second derivatives of $f(\mathbf{p}|\Gamma,\Lambda)$ given Eq. 11 with respect to the $k=5$ parameters and evaluating the resulting $5 \times 5$ matrix at the optimal parameter values. Because the first level optimization for $\hat{\mathbf{p}}$ is very fast, we perform the second level optimization by evaluating $\log(Z(\Gamma,\Lambda))$ on a two dimensional mesh and choosing the arguments, $(\hat{\Gamma},\hat{\Lambda})$ that yield the maximum (Fig. 2). The two level evidence maximization algorithm allows for the automatic callibration of the relative importance of the likelihood and prior for the determination of the posterior probability for the parameters, which is then used to find the most probable parameter values along with their errorbars. Our best estimate for the parameters are given by the $\hat{\mathbf{p}}$ that maximizes $f(\mathbf{p}|\hat{\Gamma},\hat{\Lambda})$. The error bars for the parameters are determined using Laplace's approximation (e.g. Teng et al., 2014).

## 3 Results and discussion

### 3.1 Model performance

A Bayesian inverse method was previously applied to MedFlux IRS SV sediment trap data to estimate particle exchange parameters (Wang et al., 2017). In that study, the method was tested by creating a set of synthetic data using a finite difference model with a group of "true parameters", then contaminating the data with random errors, finally recovering "true parameters" by using the inverse method. In this study, the same mathematical model is applied; however, instead of using the "two-layer" model created to model data collected using SV sediment traps, we use a box-model to model the data collected using large volume pumps, a sample collection method that is more common in oceanographic surveys than SV sediment traps.



One significant difference between pump and SV sediment trap data is that pumps measure concentrations and SV sediment traps measure particle flux at different sinking velocities. The previous "two-layer" model is tailored for flux data because flux differences between trap depths are known. In addition, because the data are available at only three trap depths, they are inadequate to build a traditional box model (Wang et al., 2016, 2017). In contrast, pump data that are available at 6-7 depths

enable us to create a classic box model in which the upper box's output becomes the input to the lower box.

A comparison of model versus observational data and contour plots used to select relative strengths of parameter and data constraint factors are shown in Fig. 2. As can be seen, though the model generally underestimates the observation, considering the highly variable data (up to 7 orders of magnitudes), the inverse model does a decent job of recovering the observational data ($R^2 = 0.88$).

Note that the uppermost samples are not included in the fits because they are used as the boundary conditions for the models. To test the sensitivity of the results to the boundary conditions, we increased or decreased the upper most pigment and POC concentrations by 20%. The results are shown in Table 2 and Fig.3. As can be seen, change of boundary conditions does not significantly change optimal parameters, except for disaggregation rate constants that changed by ∼13 and ∼12% when the boundary condition increased and decreased, respectively. As can be seen from Fig.3, the correlation between model and

observation data does not change when the boundary condition is changed.

### 3.2   Particle sinking velocity

In contrast to previous box models that assign an arbitrary sinking velocity for large particles (Clegg and Whitfield, 1991; Murnane et al., 1990; Murnane, 1994; Cochran et al., 1993; Cochran et al., 2000), depth dependent sinking velocity in this study is built by using a vertical flux transport operator **S**. Variable sinking velocities have been shown to better represent

sinking particles in geochemical models (Kriest and Oschlies, 2008). The power law for POM remineralization with depth, known as the Martin Curve, indicates either increasing sinking speed or decreasing remineralization rate with depth (Martin et al., 1987). Increasing sinking speed has also been suggested by analyses of sediment trap array data (Berelson, 2002).

However, Xue and Armstrong (2009) applied the "benchmark" method to IRS time series sediment trap data collected at multiple locations, they obtained an average sinking speed of 220±65 m/d, and concluded that particle sinking speed did

not change with depth. Here we do not apply a constant sinking speed for three reasons. First, results based on sediment trap data may not be applicable to the large-volume pump data. The relatively short sampling time that the pump is deployed decreases the chance of capturing really fast sinking particles, which are rare in the ocean. Traps on the other hand are deployed for months, and may thus capture a larger amount of fast sinking particles. We calculated particle sinking velocity based on optimal parameters. Large-sized particle sinking velocity ranges from  8 to  212 m/d with an average of 77.63±79.69 m/d.

Both the range and mean SV are consistent with previous estimates of aggregates(Alldredge and Gotschalk, 1988; Asper et al., 1992; Pilskaln et al., 1998), but lower than that of fecal pellets (Fowler and Small, 1972; Komar et al., 1981). Second, without a rate-determining factor like decay half-life in thorium models, the data cannot simultaneously determine sinking speed and





particle exchange rates. We tested our model by applying two different sinking speeds: 100 m/d and 200 m/d. As shown in Table 3, the particle remineralization rate constant and disaggregation rate constant change proportionally with sinking speed. Lastly but most importantly, our objective here is to introduce a versatile method for calibrating parameters and to test the sensitivity of these inferences to different sampling methods (SV sediment traps versus large volume pumps) and to different

tracers (thorium versus pigments). Since particle sinking velocities (SV) are rarely precisely measured, the arbitrary assignment (e.g. 100 or 200 m/d ) of a sinking velocity could cause large uncertainties (Table 3). Our method does not require a value for SV (but can switch to constant SV mode if available SV data are reliable), whereas, it optimizes the Martin Curve exponential *b*, which is more commonly used than SV.

### 3.3    Particle exchange rate constants

The inverse model predicts a small-particle POC remineralization rate constant of 1.25 yr$^{-1}$ (0.80 yr). Chl-*a* degradation and phaeopigment degradation rate constants are not well constrained by the data, and are mostly determined by the prior values. Therefore, we adopted the values of these two parameters from Wang et al. (2017), although that work was for sediment trap material. The small-sized POC remineralization rate constant estimated here is highly comparable to the previous estimate (1.5 yr$^{-1}$ for slow-sinking particles) using SV sediment trap data from this site (Wang et al., 2017). The POC remineralization rate

constant estimated here is lower than that estimated in other regions at a corresponding depth range (Clegg et al., 1991; Clegg and Whitfield, 1991). Geographical differences could account for this.

The aggregation rate constants are surprisingly consistent (7.65 yr$^{-1}$ in this study, 3.20 yr$^{-1}$ in (Wang et al., 2017)) considering such different sampling techniques. The disaggregation rate constant estimated in this study (106.09 yr$^{-1}$) is $\sim 30\%$ lower than that in (Wang et al., 2017) (149.9 yr$^{-1}$); both fall within the range of previous estimates (Murnane et al., 1996;

Lerner et al., 2016). In addition, aggregation and disaggregation rate constants in both studies were consistent with the idea that small and large particles exchange their components via aggregation and disaggregation during low flux times. Abramson et al. (2010) suggested this earlier based on a simple, but non-quantitative, comparison of the choropigment composition of traps and pumps at this site in both 2003 and 2005. They suggested that aggregation and disaggregation were low during high flux times, and high during low flux times.

However, when compared to results modeled using thorium data that were collected using SV sediment traps at the same location and the same time (Wang et al., 2016), aggregation and disaggregation rate constants are higher when pigment data are used (Wang et al. (2017) and this study). These comparisons are intriguing because they suggest that parameters estimated using the same tracers but different sampling techniques were more consistent than that using the same techniques but different tracers.



## 4 Conclusions

In this study, we applied Bayesian inverse methods to the data sampled using large volume pumps at the Mediterranean MedFlux site. The samples used were collected in a low flux period well after the normal spring bloom. Particles experienced extensive aggregation and disaggregation. Comparisons to previous results estimated using thorium and pigment data but sampled using sediment traps indicate that the two different sampling and model methods have less influence on particle aggregation and disaggregation rate constant estimations than do the two different tracers themselves. The comparisons also highlight the different characteristics between pigment tracers and thorium tracers.

*Competing interests.* The authors declare that they have no conflict of interests

*Acknowledgements.* We sincerely thank all our MedFlux colleagues, especially J. Kirk Cochran and Juan-Carlos Miquel for samples from their in-situ pumps and the captain and crew of the R/V Endeavor. We appreciate the financial support provided by the Chemical Oceanography Program of the US National Science Foundation (MedFlux 01-36370 and 06-22754 and BarFlux 10-61128), and the Division of Ocean Sciences of the US National Science Foundation (OCE-1436922). The International Atomic Energy Agency (IAEA) supported the deployment of MEL in-situ pumps.



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



**Table 1.** Concentrations in $\mu$M of data collected using large volume pumps at the French DYFAMED Site (4320'N, 740'E) in the Mediterranean Sea in May, 2005 (Cochran et al., 2009). The left-side columns are for small particles (1-70 $\mu$m). The rest are for large particles (>70 $\mu$m). According to Abramson et al. (2010), replicate pigment samples (i.e., duplicate punches taken from different places on the same pump filter) generally differed within 30% (calculated by propagation of error). POC data have much lower uncertainty (2%). In the model, we use 30% uncertainty for pigments, and 2% for POC.

| Site | Depth | Chl $a$ | POC | $\sum(Pheo)$ | Chl $a$ | POC | $\sum(Pheo)$ |
|------|-------|---------|-----|--------------|---------|-----|--------------|
|      | (m)   | ($\mu$M) | ($\mu$M) | ($\mu$M) | ($\mu$M) | ($\mu$M) | ($\mu$M) |
| D | 100 | $1.30\times10^{-5}$ | 1.89 | $3.32\times10^{-5}$ | $1.76\times10^{-8}$ | $2.32\times10^{-1}$ | $2.34\times10^{-7}$ |
| Y | 125 | $9.35\times10^{-6}$ | 1.26 | $1.49\times10^{-5}$ | $1.63\times10^{-8}$ | $1.02\times10^{-1}$ | $1.01\times10^{-7}$ |
| F | 150 | $6.21\times10^{-6}$ | 1.04 | $6.52\times10^{-6}$ | $1.98\times10^{-8}$ | $8.23\times10^{-2}$ | $1.41\times10^{-7}$ |
| A | 200 | $2.63\times10^{-6}$ | 0.64 | $6.60\times10^{-6}$ | $3.13\times10^{-8}$ | $1.56\times10^{-1}$ | $1.15\times10^{-7}$ |
| M | 300 | $1.37\times10^{-6}$ | 0.44 | $1.88\times10^{-6}$ | $1.22\times10^{-8}$ | $4.15\times10^{-2}$ | $8.91\times10^{-8}$ |
| E | 500 | $1.54\times10^{-6}$ | 0.44 | $1.81\times10^{-6}$ | $1.36\times10^{-8}$ | $8.12\times10^{-2}$ | $1.22\times10^{-7}$ |
| D | 750 | $3.75\times10^{-7}$ | 0.51 | $1.12\times10^{-6}$ | $3.43\times10^{-9}$ | $6.44\times10^{-2}$ | $6.41\times10^{-8}$ |

Sum(Pheo) means $\sum(pheophorbide, pyropheophorbide, pheophytin)$





**Table 2.** Particle aggregation, disaggregation, chloropiment degradation, POC dissolution rate constants estimated based on pump data sampled at DYFAMED site (unit: $yr^{-1}$ except for Martin curve exponential $b$ that is dimensionless). Pigment remineralization rate constants ($d_1$ and $d_3$) are from (Wang et al., 2017), due to the fact that the model and data are unable to constrain these two parameters. $\exp(\mathbf{p_0})$ is parameter prior, and $\sigma_p^2$ is variance. The last two rows $\eta = 1.2$ and $\eta = 0.8$ are sensitivity tests for boundary conditions.

|  | $b$ | $d_1$ | $d_2$ | $d_3$ | $\beta$ | $\beta_{-1}$ |
|---|---|---|---|---|---|---|
| $\exp(\mathbf{p_0})$ | 0.87 | 1.6 | 0.32 | 2.1 | 5.47 | 148.41 |
| $\sigma_p^2$ | 0.16 | - | 2.98 | - | 4.01 | 6.64 |
| Optimal | $0.89^{+0.08}_{-0.07}$ | - | $1.25^{+0.55}_{-0.38}$ | - | $7.65^{+3.35}_{-2.33}$ | $106.09^{+39.13}_{-28.59}$ |
| $\eta = 1.2$ | $0.89^{+0.12}_{-0.11}$ | - | $1.29^{+0.88}_{-0.52}$ | - | $7.95^{+5.36}_{-3.20}$ | $93.42^{+51.64}_{-33.26}$ |
| $\eta = 0.8$ | $0.89^{+0.05}_{-0.04}$ | - | $1.29^{+0.31}_{-0.25}$ | - | $6.77^{+1.68}_{-1.35}$ | $120.69^{+26.39}_{-21.65}$ |



**Table 3.** Parameter values of constant sinking speed model (unit: $\mathrm{yr}^{-1}$ except for sinking speed (SV) that is m/d). Parameters are defined in Fig.1.

| SV | $d_1$ | $d_2$ | $d_3$ | $\beta$ | $\beta_{-1}$ |
|---|---|---|---|---|---|
| $\omega = 100$ | 0.03 | 16.75 | 1.34 | 0.16 | 98.87 |
| $\omega = 200$ | 0.04 | 33.49 | 2.64 | 0.34 | 197.76 |




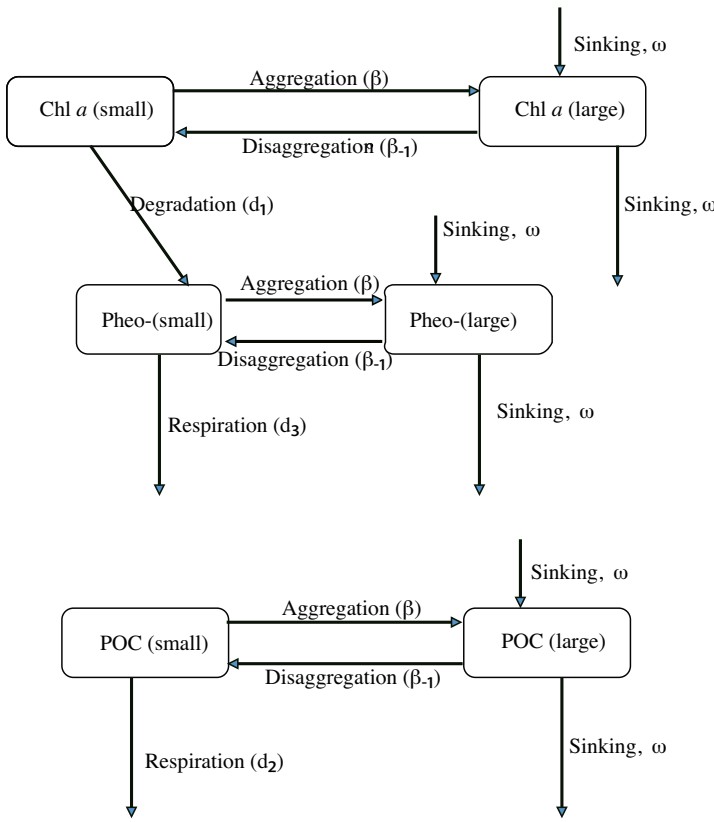

**Figure 1.** Conceptual model modified from the one used in Wang et al. (2017). One significant difference is that small and large size particles instead of slow- and fast-sinking particles are used in this study. Another difference is that small particles are assumed not sinking. In addition, aggregation and disaggregation occur between small and large particles, instead of between slow- and fast- sinking particles as in Wang et al. (2017)





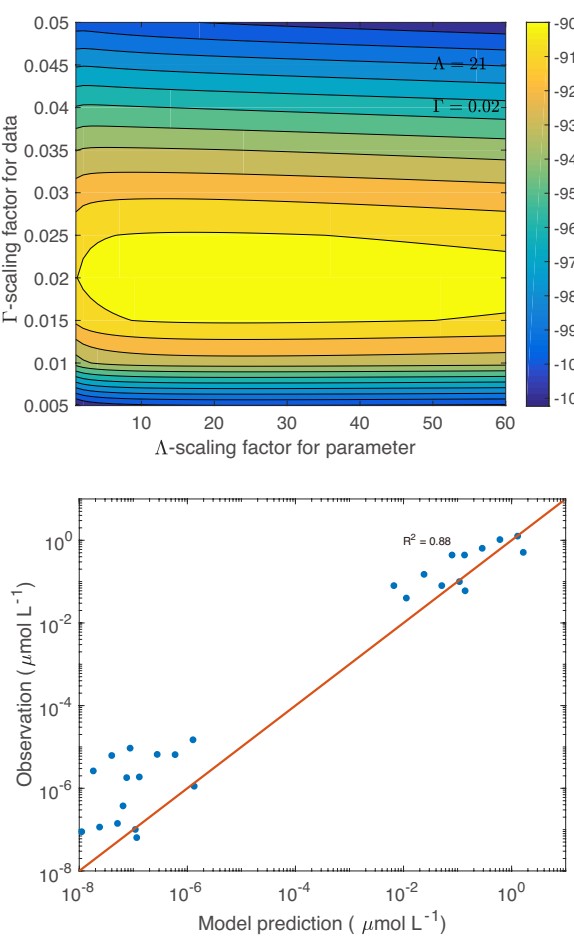

**Figure 2.** Contour plots of the logarithm of the evidence along with model versus observation comparisons. The maximum Λ and Γ values are added in the figure.





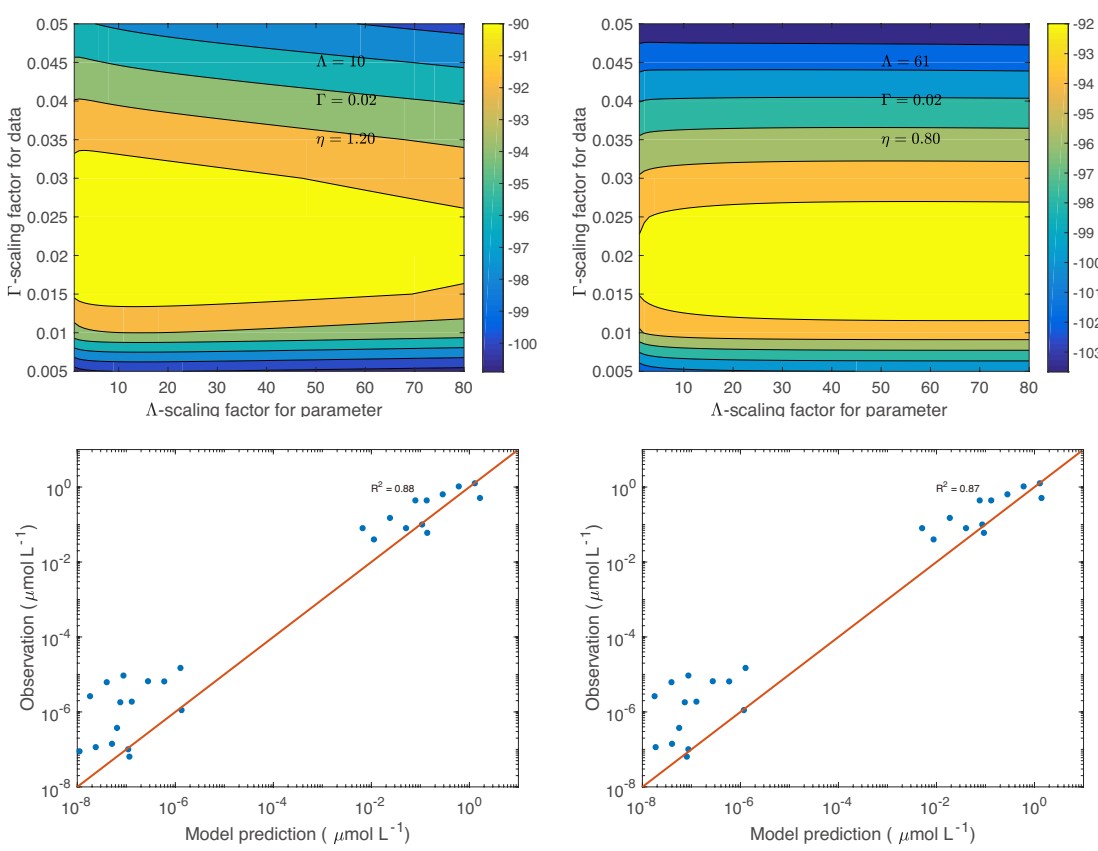

**Figure 3.** Contour plots of the logarithm of the evidence along with model versus observation comparisons, when boundary concentrations increase (left panel) and decrease (right panel) by 20%.