# Peer review of "Inferring particle dynamics in the Mediterranean Sea from in-situ pump POC and chloropigment data using Bayesian statistics"

_Biogeosciences, 2018_

## Referee Comment (RC1) · Anonymous Referee #1 · 3 Apr 2018

**To the authors (Wang et al.)**

The authors aim at interpreting in-situ pump data from the DYFAMED station in the Mediterranean Sea. For this purpose they propose a model including aggregation and disaggregation of particles of two size classes, respiration, degradation and sinking. Data and model values are considered at 7 depth levels. The model parameters are estimated in two steps by an Bayesian approach.

At the time of sampling (May 2005) the particle numbers in the water column are very low and thus I doubt that aggregation would have played a significant role in particle dynamics. Aggregation of particles is an essentially non-linear process (because particles have to meet in order to aggregate) and thus in my opinion a linear term for the description of aggregation is highly inappropriate. Further, the authors make a steady-state assumption without justification or further discussion. Application of the estimated ('optimal') parameters in the various model equations yields no convincing support for the steady-state assumption. The whole text seems to be largely driven by methodological aspects (Bayesian approach) and neglects a discussion of the data and the estimated model parameters (are the estimates plausible/do they make sense?). Given the deficits in the model formulation ('linear aggregation', 'steady-state') and the question whether the data used contain valid information about the processes included in the model, I could not see what a reader can learn from this investigation and thus I do not recommend publication in Biogeosciences.

Further remarks:

Abstract: "The estimated aggregation and disaggregation rate constants were $7.65 + 3.35 - 2.33$ (0.13 yr) and $106.09 + 39.13 - 28.59$ yr$^{-1}$ (0.01 yr), respectively, which indicates that particle aggregation and disaggregation were extensive at the studied depths (125-750 m) in May after the spring bloom had ended and flux was low."
Disaggregation is faster (lower time constant: 0.01 yr) than aggregation (time constant 0.13 yr) and thus one would expect less and less aggregates.

p. 1 L19 Stoke's law $\Rightarrow$ Stokes' law
(named after George Gabriel Stokes)

p. 2 L5-7 ”3) dissolved thorium activity is orders of magnitude higher than particulate activity, thus in models, dissolved thorium influences the adsorption-desorption balance more than particulate thorium does (Wang et al., 2016).”
Not clear to me: 'What is the disadvantage here?' A problem for models only or also for the real world?

p. 2 L16 ”Wang et al. (2017) compared rate constants ...”
Rate constants: for which process(es)?

p. 2 L18-20 ”There were clear differences: aggregation and disaggregation rate constants estimated from chloropigments were orders of magnitude higher than those from thorium tracers. These results were thought to be due to the differences in the way thorium and chlorophyll exist on and within particles.”
Not clear, what authors like to tell us.

p. 2 L26-29 this should be dropped: ”... since traditionally it is thought that aggregation is the process of small particles colliding with other particles and becoming larger, and disaggregation is the process of larger particles breaking up into smaller particles. Whereas, particles in fast sinking categories are a combination of large and small particles, because some small particles have high sinking speed, and vice versa ...”

p. 3 L12-13 ”These inter-comparisons enabled us to investigate whether sampling techniques or the tracer itself imposed a stronger constraint on modeled rate constants.” ???

p. 3 L17 (4320'N, 740'E) $\Rightarrow$ (43°20' N, 7°40' E)

p. 4 L16 ”If we discrete ...” $\Rightarrow$ ”If we divide ...”

p. 4 L27-28 ”We assume first-order reaction kinetics for aggregation and disaggregation, which is a gross simplification since the real reaction kinetics is not known.”
If the real reaction kinetics is not known, then it is not known whether first-order reaction kinetics is a gross simplification.
The kinetics of aggregation of biogenic marine particles has been described by Jackson (1990) and many others: it is essentially non-linear (because particle have to 'meet each other' in order to aggregate). The kinetics depends on size, sinking speed, stickiness of particles (for details compare, for example, Jackson, 1990).

p. 5 L1 "We thus use the same mathematical method and conceptual model found in other published studies ..."
Used methods (including first-order reaction kinetics for aggregation) have to be justified based on theories (first principles) and/or observations; reference to 'other published studies' is not enough.

p. 5-6 "Here we assume the system is at steady state, just as has been done in Wang et al. (2016, 2017) and almost all similar work, e.g.(Murnane et al., 1990; Clegg and Whitfield, 1991; Marchal and Lam, 2012; Lerner et al., 2016, 2017), to make the results comparable."
The steady state assumption is not sufficiently justified (again: reference to 'similar work' is not enough).

p. 7 L6 "hyperv paramter" $\Rightarrow$ "hyper parameter"

p. 7 L7-8 "Typically, we obtain the optimal parameters in less than 50 iterations." 'Iterations' is mentioned here for the first time. What is iterated?

Table 2: typo: $\beta \Rightarrow \beta_1$

**References**

[1] Jackson, G.A. A model of the formation of marine algal flocs by physical coagulation processes. *Deep-Sea Res.*, 37(8):1197–1211, 1990.

---

## Author Comment (AC1) · 4 Apr 2018

**Reply to Anon. Rev. 1 (Wang et al.)**

Reviewer: The authors aim at interpreting in-situ pump data from the DYFAMED station in the Mediterranean Sea. For this purpose they propose a model including aggregation and disaggregation of particles of two size classes, respiration, degradation and sinking. Data and model values are considered at 7 depth levels. The model parameters are estimated in two steps by an Bayesian approach.

At the time of sampling (May 2005) the particle numbers in the water column are very low and thus I doubt that aggregation would have played a significant role in particle dynamics. Aggregation of particles is an essentially non-linear process (because particles have to meet in order to aggregate) and thus in my opinion a linear term for the description of aggregation is highly inappropriate. Further, the authors make a steady-state assumption without justification or further discussion. Application of the estimated ('optimal') parameters in the various model equations yields no convincing support for the steady-state assumption. The whole text seems to be largely driven by methodological aspects (Bayesian approach) and neglects a discussion of the data and the estimated model parameters (are the estimates plausible/do they make sense?). Given the deficits in the model formulation ('linear aggregation', 'steady-state') and the question whether the data used contain valid information about the processes included in the model, I could not see what a reader can learn from this investigation and thus I do not recommend publication in Biogeosciences.

Our reply: As we stated on p. 3 lines 8-13, the objectives of the study were 1) to introduce a new method for using pump collected chloropigment tracers to infer particle dynamics rate constants, and 2) to compare to previous studies at the same site, and thus, to infer if sampling methods or tracers themselves have more influence on particle aggregation and disaggregation rate constant estimates. Therefore, "The whole text seems to be largely driven by methodological aspects" is not a deficit; developing this method is what the study aimed to do. From our study, readers can learn 1) a new approach to determining rate constants, and 2) a proof that if the same tracers are used in the model, sampling methods have little influence on rate constant estimates.

The reviewer does not believe that the "linear aggregation" and "steady state" assumptions are valid. First, we argue that linear aggregation is not without merit, especially when aggregation is between the same kinds of particles. As the reviewer stated in his/her comments, "particles have to meet in order to aggregate". Therefore aggregation has to be related in some way to the concentration of particles, no matter whether the particle concentration is high or low. By assuming a first-order reaction, we have

$$\text{aggregation} = \alpha[P],$$

where $\alpha$ is an aggregation rate constant, and $[P]$ is the particle concentration. Higher particle concentration results in a higher chance to collide, and thus more aggregation. In most empirical studies, including with the current data, we have no way to build and test a complicated model such as that in Jackson (1990). Also, unlike in Jackson (1990),

we are dealing with particles below the euphotic depth, so do not consider algal division and other euphotic zone processes. Many previous models have assumed first-order reaction kinetics and show reasonable results (Murnane et al., 1990; Clegg and Whitfield, 1991; Marchal and Lam, 2012; Lerner et al., 2016, 2017).

Second, we assume "steady state" since we are using only one concentration profile. We do not have enough information to build a non-steady-state model. To achieve our second goal, to compare the influence of sampling techniques and tracers on parameter estimations, we have to make the same assumptions as we did in previous studies (Wang et al., 2016; 2017).

Reviewer: Further remarks:
Abstract: "The estimated aggregation and disaggregation rate constants were 7.65 + 3.35 − 2.33 (0.13 yr) and 106.09 + 39.13 − 28.59 yr$^{-1}$ (0.01 yr), respectively, which indicates that particle aggregation and disaggregation were extensive at the studied depths (125-750 m) in May after the spring bloom had ended and flux was low." Disaggregation is faster (lower time constant: 0.01 yr) than aggregation (time constant 0.13 yr) and thus one would expect less and less aggregates.

Our reply: We do not understand the reasoning here. Lower turnover times (0.01 year for disaggregation, and 0.13 years for aggregation) indicate higher reactions. We were not comparing absolute amounts of aggregation vs disaggregation here. To determine whether more particles were disaggregating than aggregating, we would have to determine actual rates, not rate constants. That was not our purpose.

In addition, another study at the same site at the same sampling time indicates high aggregation and disaggregation using totally different empirical methods (Abramson et al., 2010).

Reviewer: p. 1 L19 Stoke's law ⇒ Stokes' law (named after George Gabriel Stokes)

Our reply: We will fix this. Thank you.

Reviewer: p. 2 L5-7 "3) dissolved thorium activity is orders of magnitude higher than particulate activity, thus in models, dissolved thorium influences the adsorption-desorption balance more than particulate thorium does (Wang et al., 2016)."
Not clear to me: 'What is the disadvantage here?' A problem for models only or also for the real world?

Our reply: This is a model problem when Th is used, as we stated in the paper.

Reviewer: p. 2 L16 "Wang et al. (2017) compared rate constants ..." Rate constants: for which process(es)?

Our reply:  Aggregation and disaggregation rate constants. We made it clear in the paper.

Reviewer: p. 2 L18-20 "There were clear differences: aggregation and disaggregation rate constants estimated from chloropigments were orders of magnitude higher than those from thorium tracers. These results were thought to be due to the differences in the way thorium and chlorophyll exist on and within particles."
Not clear, what authors like to tell us.

Our reply:  We meant that thorium and chloropigments have different properties, and trace different particle processes. We discussed this more in our previous paper (Wang et al. 2017) and should have repeated it here (and will do so). In that paper, we showed that particulate pigments cycled differently than particulate thorium. Thorium and other surface-active elements would behave very differently from chloropigments as they adsorb and desorb from the surface of the particle. Chloropigments are more likely to be an integral part of the particle.

Reviewer: p. 2 L26-29 this should be dropped: "... since traditionally it is thought that aggregation is the process of small particles colliding with other particles and becoming larger, and disaggregation is the process of larger particles breaking up into smaller particles. Whereas, particles in fast sinking categories are a combination of large and small particles, because some small particles have high sinking speed, and vice versa …"

Our reply:  Why would you like this dropped?

Reviewer: p. 3 L12-13 "These inter-comparisons enabled us to investigate whether sampling techniques or the tracer itself imposed a stronger constraint on modeled rate constants." ???

Our reply: We assume you mean that he sentence was unclear? How about:  "The inter-comparisons we describe here allowed us to investigate whether sampling techniques (pumps vs. traps) or the tracer itself (chloropigments vs. Th) imposed the stronger constraint on modeled rate constants."

Reviewer: p. 3 L17 (4320'N, 740'E) $\Rightarrow$ (43$^\circ$ 20' N, 7$^\circ$ 40' E)

Our reply:  We will fix this.

Reviewer: p. 4 L16 "If we discrete ..." ⇒ "If we divide …"

Our reply:  We will fix this.

Reviewer: p. 4 L27-28 "We assume first-order reaction kinetics for aggregation and disaggregation, which is a gross simplification since the real reaction kinetics is not known."
If the real reaction kinetics is not known, then it is not known whether first-order reaction kinetics is a gross simplification.
You are correct.  We will change this to "which is probably a gross simplification since the real reaction kinetics is not known."

The kinetics of aggregation of biogenic marine particles has been described by Jackson (1990) and many others: it is essentially non-linear (because particle have to 'meet each other' in order to aggregate). The kinetics depends on size, sinking speed, stickiness of particles (for details compare, for example, Jackson, 1990).

Our reply:  In the model we use, aggregation is a process of smaller particles aggregating to form larger particles. We assume that small particles are single, non-sinking particles, and have the same stickiness. As discussed above, with the data we have, we cannot build a model like that in Jackson (1990).

Also, the first-order kinetic model is reasonable when aggregation is between the same kind of particles. By assuming first reaction kinetics, we have
$$\text{aggregation} = \alpha[P]$$
Thus, high particle concentration indicates high chance of particle collision, thus high particle aggregation. Previous model studies assuming first reaction kinetics also come to reasonable results (Murnane et al., 1990; Clegg and Whitfield, 1991; Marchal and Lam, 2012; Lerner et al., 2016, 2017).

Reviewer: p. 5 L1 "We thus use the same mathematical method and conceptual model found in other published studies ..."
Used methods (including first-order reaction kinetics for aggregation) have to be justified based on theories (first principles) and/or observations; reference to 'other published studies' is not enough.

Our reply:  With measurements at only one depth profile. We have no way to build a non-steady-state model. And for the purpose of model comparison, similar steady state assumptions as in previous studies (Wang et al 2016; 2017) were made.  We use the same justifications as in those papers, and do not presume to repeat them all.

Reviewer: p. 5-6 "Here we assume the system is at steady state, just as has been done in Wang et al. (2016, 2017) and almost all similar work, e.g.(Murnane et al., 1990; Clegg

and Whitfield, 1991; Marchal and Lam, 2012; Lerner et al., 2016, 2017), to make the results comparable."
The steady state assumption is not sufficiently justified (again: reference to 'similar work' is not enough).

Our reply: See above reply.

Reviewer: p. 7 L6 "hyperv paramter" ⇒ "hyper parameter"

Our reply: We will fix this.

Reviewer: p. 7 L7-8 "Typically, we obtain the optimal parameters in less than 50 iterations." 'Iterations' is mentioned here for the first time. What is iterated?

Our reply: The model iteratively searches for optimal parameters.

Reviewer: Table 2: typo: $\beta \Rightarrow \beta_1$

Our reply: We will fix this.

**Reviewer: References**

[1] Jackson, G.A. A model of the formation of marine algal flocs by physical coagulation processes. Deep-Sea Res., 37(8):1197–1211, 1990.

References mentioned above

Abramson, L., Lee, C., Liu, Z., Wakeham, S., and Szlosek, J.: Exchange between suspended and sinking particles in the northwest Mediterranean as inferred from the organic composition of in situ pump and sediment trap samples, Limnol. Oceanogr., 55, 725–739, 2010.

Clegg, S. L. and Whitfield, M.: A generalized model for the scavenging of trace metals in the open ocean-II. Thorium scavenging, Deep-Sea Res. I, 38, 91–120, 1991.

Jackson, G.A. A model of the formation of marine algal flocs by physical coagulation processes. Deep-Sea Res., 37(8):1197–1211, 1990.

Lerner, P., Marchal, O., Lam, P. J., Anderson, R. F., Buesseler, K., Charette, M. A., Edwards, R. L., Hayes, C. T., Huang, K. F., Lu, Y., Robinson, L. F., and Solow, A.: Testing models of thorium and particle cycling in the ocean using data from station GT11-22 of the U.S. GEOTRACES North Atlantic section, Deep-Sea. Res. I, 113, 57–79, 2016.

Lerner, P., Marchal, O., Lam, P. J., Buesseler, K., and Charette, M.: Kinetics of thorium and particle cycling along the US GEOTRACES North Atlantic Transect, Deep Sea Research Part I: Oceanographic Research Papers, 125, 106–128, 2017.

Marchal, O. and Lam, P. J.: What can paired measurements of Th isotope activity and particle concentration tell us about particle cycling in the ocean?, Geochim. Cosmochim. Acta, 90, 126–148, 2012.

Murnane, R. J., Sarmiento, J. L., and Bacon, M. P.: Thorium isotopes, particle cycling models, and inverse calculations of model rate constants, J. Geophys. Res., 95, 16 195, 1990.

Wang, W.-L., Armstrong, R. A., Cochran, J. K., and Heilbrun, C.: 230Th and 234Th as coupled tracers of particle cycling in the ocean: A maximum likelihood approach, Deep Sea Res. I, 111, 61–70, 2016.

Wang, W.-L., Lee, C., Cochran, J. K., Primeau, F. W., and Armstrong, R. A.: A novel statistical analysis of chloropigment fluxes to constrain particle exchange and organic matter remineralization rate constants in the Mediterranean Sea, Mar. Chem., 2017.

---

## Referee Comment (RC2) · Anonymous Referee #2 · 7 May 2018

Review of Wang et al: This paper uses a Bayesian approach to determine rate parameters for a simple particle model using data collected during the MEDFLUX program at the DYFAMED site in the Mediterranean Sea. I think that the authors have a potentially useful approach here, but the presentation of the manuscript and their ideas requires a lot more work.

P1. Line 18: The authors state that "Particle density and size determine particle sinking speed". This is a very contentious statement. This problem has a very long history, and to date, no relationship between size and settling velocity has even been shown. Density does seem to be a determining factor, but size, not so much.

P1. Lines 21–22: Lots of work was done historically on this problem using isotopes, starting with the work of Bacon and Anderson, and Clegg and Whitfield which the authors cite later, but should also be cited here.

P1. Line 23: The authors state "...much of what is currently known about these processes is from work with particulate thorium." This is a gross simplification and does not represent the myriad techniques that have been and are used to help us understand particle process in the ocean. I would argue that use of thorium tracers is one tool that has been used, but a considerable amount of what we know about aggregation and disaggregation has come from other techniques including optical methods, staining methods, laboratory experimental methods (e.g. rolling tanks), and modeling. Thorium tells us very little about the biological processes that enhance aggregation and disaggregation, or determine the strength of particles. So whilst use of thorium isotopes is a critically important, it is only one of many techniques. One could equally make the case for optical techniques being the main source of information.

P4. Lines 17–18: The authors make two very fundamental, connected assumptions: that sinking speed increases linearly with depth and that the flux attenuation profile is a power law. However, as the authors state on page 8, other observations at the MEDFLUX site show that sinking speeds do not increase with depth. The authors explanation of why they make this assumption does not seem to make sense to me. If you are applying an inverse model to data at a given site, you shouldn't make an assumption that clearly does not hold at that site, unless you have evidence that previous estimates were in error.

Equations 1–6: The authors assume first-order reaction kinetics for aggregation and disaggregation. This is known to be incorrect — aggregation is a fundamentally non-linear process and to assume that it is a linear process depending only on the particle concentration is unphysical. Disaggregation is also not a linear process but depends on environmental process such as turbulence, or factors such as animal abundance. So, the model used by the authors is inherently unrealistic and unphysical from the

start – to see this, just analytically solve the linear odes with only the aggregation and disaggregation terms and you'll get completely unphysical solutions.

P5: Lines 6–7. The authors assume that chlorophyll is found only in the small particles, and that any chlorophyll found in larger particles comes from aggregation of small particles. If I'm reading the paper correctly, the authors use a size of 70 $\mu$m to separate large and small particles. So in this model, there is no photosynthesis in particles greater than 70 $\mu$m? This rules out most diatoms and other large phytoplankton. This surely cannot be correct.

P5: Line 20: I must be missing something here. The authors state that the transpose of the vector c is has 48 components, but only 6 are listed.

P6: Lines 5 to end of section: This is very unclear. Why use a Bayesian approach? What do we gain from this? Why won't a more standard approach also work. I'm not averse to using Bayesian approaches, and they are often more informative and successful than standard frequentist approaches. But it's unclear to me why they should be used here. What is more, the explanation of the technique given here is unclear – why are two optimizations needed? Why do we need to scale the data and prior precisions? Won't using the logarithms of the parameters bias the end result because you've inherently changed the statistical distribution of parameter uncertainties? (this is similar to the problems incurred by fitting a straight line to log-transformed data that obey a power law or exponential distribution). Also, the authors assume that errors are independent (in order to make their likelihood matrix diagonal). What is the justification for this? Given the data being used, I would have thought that the uncertainties were highly correlated. This whole section needs to be thought out more carefully, and be re-written to be more explanatory.

Table 1: There are no uncertainties in this table. Even if the observation uncertainties are estimated by the limitations/sensitivity of the instruments/methodologies, they should be given! What is more, taking these uncertainties into account will affect the

uncertainties in the parameter estimates given in Table 2.

Figures 2 and 3: Perhaps I missed this, but there seems to be no significant discussion of the fact that their model consistently under-predicts the observations. The quoted R-squared value is obviously being driven by the two clusters of data. This needs to be examined and discussed in detail.

————————————————

---

## Author Comment (AC2) · 10 May 2018

**Suggestions for technical corrections or reasons for rejection**

Review of Wang et al:
This paper uses a Bayesian approach to determine rate parameters for a simple particle model using data collected during the MEDFLUX program at the DYFAMED site in the Mediterranean Sea. I think that the authors have a potentially useful approach here, but the presentation of the manuscript and their ideas requires a lot more work.

P1. Line 18: The authors state that "Particle density and size determine particle sinking speed". This is a very contentious statement. This problem has a very long history, and to date, no relationship between size and settling velocity has even been shown. Density does seem to be a determining factor, but size, not so much.

OUR RESPONSE: According to Stoke's law, particle sinking velocity ($\omega_{sinkk}$) is proportional to the square of equivalent particle radius $r$,

$$\omega_{sinkk} = \frac{2g \cdot r^2 \cdot (\rho_1 - \rho_2)}{9\mu},$$

where $\rho_1$ is particle density, and $\rho_2$ is the density of liquid in which particles sink. $g, \mu$ are gravitational acceleration and liquid viscosity, respectively. Clegg and Whitfield (1991) compiled particle sinking velocity data. When plotted on double logarithm scale, particle sinking velocity generally increased with particle size. On the other hand, recent studies (e.g. McDonnell and Bensseler 2010) show that smaller particles can have higher sinking velocity than some larger particles. This is also why we have been advocating the use of sinking velocity (SV) sediment traps, which can measure in situ particle sinking velocity, to constrain particle dynamics. However, as we mentioned in the manuscript, SV sediment traps have not been widely used due to the lack of commercial availability. This is also why we conducted the submitted study. Our main objective here is to compare how different sampling methods can influence parameter estimates.

Changed to: "Theoretically, particle density and size determine particle sinking speed according to Stoke's Law, and thus residence time in the water column (McCave, 1975; Armstrong et al., 2002). However, processes such as aggregation and disaggregation can alter this relationship and ultimately influence particle transfer efficiency (e.g. McDonnell and Buesseler 2010.) "

P1. Lines 21-22: Lots of work was done historically on this problem using isotopes, starting with the work of Bacon and Anderson, and Clegg and Whitfield which the authors cite later, but should also be cited here.

OUR RESPONSE: citations have been added.

P1. Line 23: The authors state "…much of what is currently known about these processes is from work with particulate thorium." This is a gross simplification and does not represent the myriad techniques that have been and are used to help us understand particle process in the ocean. I would argue that use of thorium tracers is one tool that has been used, but a considerable amount of what we know about aggregation and disaggregation has come from other techniques including optical methods, staining methods, laboratory experimental methods (e.g. rolling tanks), and modeling. Thorium tells us very little about the biological processes that enhance aggregation and disaggregation, or determine the strength of particles. So whilst use of thorium isotopes is a critically important, it is only one of many techniques. One could equally make the case for optical techniques being the main source of information.

OUR RESPONSE:  We did not mean to ignore all the considerable work done on aggregation and disaggregation by other techniques, but to specifically focus on the question of quantitative dynamics.

Changed to: The quantitative dynamics of particle aggregation and disaggregation processes has been studied using thorium isotopes (Murnane, 1994; Murnane et al., 1996; Cochran et al., 1993; Cochran et al., 2000), and much of what we currently know about the dynamics of these processes is from work with particulate thorium.

P4. Lines 17-18: The authors make two very fundamental, connected assumptions: that sinking speed increases linearly with depth and that the flux attenuation profile is a power law. However, as the authors state on page 8, other observations at the MEDFLUX site show that sinking speeds do not increase with depth. The authors explanation of why they make this assumption does not seem to make sense to me. If you are applying an inverse model to data at a given site, you shouldn't make an assumption that clearly does not hold at that site, unless you have evidence that previous estimates were in error.

OUR RESPONSE: One objective of the paper is to introduce a method that anyone can use in their own research. Particle sinking velocities (SV) are rarely precisely measured, and arbitrary assignment of SV can cause large uncertainties (Table 3). Our method does not ask for a SV (but can switch to constant SV mode if available SV data is reliable). Our method optimizes the Martin Curve exponential b, which is more commonly used than SV. More importantly, comparison of Table 2 and 3 shows that the aggregation and disaggregation rate constants estimated based on the constant and variable sinking speeds are comparable. Therefore, both models lead to the same conclusion that if the same tracers are used, different sampling methods do not influence aggregation and disaggregation rate estimates.

We added: First, results based on sediment trap data may not be applicable to the large-volume pump data. The relatively short sampling time that the pump is deployed decreases the chance of capturing really fast sinking particles, which are rare in the ocean. Traps on the other hand are deployed for months, and may thus capture more fast sinking particles. We calculated particle sinking velocity based on optimal parameters. We assumed that large-sized particle sinking velocity ranges from 8 to 212 m/d with an average of 77.63±79.69 m/d. Both the range and mean SV are consistent with previous estimates of aggregates (Alldredge and Gotschalk, 1988; Asper et al., 1992; Pilskaln et al., 1998), but lower than that of fecal pellets (Fowler and Small, 1972; Komar et al., 1981). Second, without a rate-determining factor like decay half-life in thorium models, the data cannot simultaneously determine sinking speed and particle exchange rates. We tested our model by applying two different sinking speeds: 100 m/d and 200 m/d. As shown in Table 3, the particle remineralization rate constant and disaggregation rate constant change proportionally with sinking speed. Lastly but most importantly, our objective here is to introduce a versatile method for calibrating parameters and to test the sensitivity of these inferences to different sampling methods (SV sediment traps versus large volume pumps) and to different tracers (thorium versus pigments). Since particle sinking velocities (SV) are rarely precisely measured, the arbitrary assignment (e.g. 100 or 200 m/d) of a sinking velocity can cause large uncertainties (Table 3). Our method does not require a value for SV (but can switch to constant SV mode if available SV data are reliable), whereas, it optimizes the Martin Curve exponential b, which is more commonly used than SV.

Equations 1-6: The authors assume first-order reaction kinetics for aggregation and disaggregation. This is known to be incorrect - aggregation is a fundamentally non-linear process and to assume that it is a linear process depending only on the particle concentration is unphysical. Disaggregation is also not a linear process but depends on environmental process such as turbulence, or factors such as animal abundance. So, the model used by the authors is inherently unrealistic and unphysical from the start - to see this, just analytically solve the linear odes with only the aggregation and disaggregation terms and you'll get completely unphysical solutions.

OUR RESPONSE: One of our objectives was to compare how different sampling methods (SV sediment trap versus large volume pump) can influence inferred particle dynamics when the same tracer is used, and to compare how different tracers (thorium versus pigments) can influence inferred particle dynamics when the same sampling method is used. To make a fair comparison with published studies (Wang et al. 2016;2017), the same mathematical method and conceptual model should be used.

We hope to try a second-order-reaction kinetic model in a separate study.

We added: We assume first-order reaction kinetics for aggregation and disaggregation, which is a gross simplification since both processes are non-linear. Our objective here, however, is to compare how different sampling methods (SV sediment trap versus large volume pump) can influence inferred

particle dynamics when the same tracer is used, and to compare how different tracers (thorium versus pigments) can influence inferred particle dynamics when the same sampling method is used. We thus use the same mathematical method and conceptual model found in other published studies (see Murnane et al. (1990); Marchal and Lam (2012); Cochran et al. (1993); Wang et al. 2016;2017).

P5: Lines 6-7. The authors assume that chlorophyll is found only in the small particles, and that any chlorophyll found in larger particles comes from aggregation of small particles. If I'm reading the paper correctly, the authors use a size of 70 µm to separate large and small particles. So in this model, there is no photosynthesis in particles greater than 70 µm? This rules out most diatoms and other large phytoplankton. This surely cannot be correct.

OUR RESPONSE: As we stated in the paper, the samples were collected in May 2005. At that time, the spring bloom was over, and diatom density should be low. Table 1 also shows that Chl-a concentrations were very low at the sampling time, indicating a low primary production condition.

We added: This is a reasonable assumption considering the sampling time (May 2005). At that time, the spring bloom was over and primary production was dominated by small phytoplankton (coccolithophorids). Table 1 also shows that Chl-a concentrations were very low at the sampling time, indicating a low primary production condition.

P5: Line 20: I must be missing something here. The authors state that the transpose of the vector c is has 48 components, but only 6 are listed.

OUR RESPONSE: POC_l POC_s … Phy_s listed in the square brackets are not scalars, each of them is a 7x1 vector. We have added further description on p. 5 and changed the notation to bold to make it clearer.

Changed to:  The bold variables such as **[POCx]**  and **[Chlx]**  (appearing in subsequent equations) represent 7 x 1 vectors, for the concentrations at each discrete depth level.

P6: Lines 5 to end of section: This is very unclear. Why use a Bayesian approach? What do we gain from this? Why won't a more standard approach also work. I'm not averse to using Bayesian approaches, and they are often more informative and successful than standard frequentist approaches. But it's unclear to me why they should be used here. What is more, the explanation of the technique given here is unclear - why are two optimizations needed? Why do we need to scale the data and prior precisions? Won't using the logarithms of the parameters bias the end result because you've inherently changed the statistical distribution of parameter uncertainties? (this is similar to the problems incurred by fitting a straight line to log-transformed data that obey a power law or exponential distribution). Also, the authors assume that errors are independent (in order to make their likelihood matrix diagonal). What is the justification for this? Given the data being used, I would have thought that the uncertainties were highly correlated. This whole section needs to be thought out more carefully, and be re-written to be more explanatory.

OUR RESPONSE: Wang et al. (2017) tested the Bayesian approach using a twin experiment. First, we created a set of synthetic data using the finite difference method, and then we contaminated the data with normal distributed error. Second, we recovered the parameters used in the finite difference model by using the Bayesian approach. The experiment shows that the Bayesian method works very well to recover parameters and estimate uncertainty.

We believe that assigning a normal prior to the logarithm of the parameters rather than to the parameters themselves provides a better reflection of our state of knowledge – all the parameters are necessarily positive, and it does not make any sense to assign prior probability to negative parameter values.

Because we have no specific information about correlations in the measurement errors we can only assume that they are independent. Doing otherwise would assume more than we know and lead to a more subjective model. It is also important to note that the assumption of independent errors in the

likelihood is conditioned on the model. The correlated uncertainties the reviewer refers to arise because s/he perceives correctly that there are mechanistic processes at work that will lead to correlations in the data. These correlations are taken into account by the mechanistic model so that the residuals become independent of each other.

With these assumptions we obtain a posterior probability that we then approximate with a normal distribution for $\log(\mathbf{p})$. The posterior probability for $\mathbf{p}$ itself does not follow normal distribution. That is why we have asymmetric error bars.

Table 1: There are no uncertainties in this table. Even if the observation uncertainties are estimated by the limitations/sensitivity of the instruments/methodologies, they should be given! What is more, taking these uncertainties into account will affect the uncertainties in the parameter estimates given in Table 2.

OUR RESPONSE: The data were obtained from an online database:
http://www.somassbu.org/research/medflux/pages/datapub/2005/In-situPumps.html
Abramson published this data earlier and said " For the concentration of each pigment and amino acid, duplicate analyses in the same samples generally differed within 10%. Replicate samples (i.e., duplicate punches taken from different places on the same pump filter) generally differed within 30% (calculated by propagation of error). Percent composition (i.e., mol % of a particular amino acid or pigment out of the total mol of all amino acids or all pigments) for each compound generally differed by < 1% between replicate samples."

We reran the model by including data uncertainties, and updated our results. The new results are almost the same as the previous ones, thus do not change our interpretation.

Figures 2 and 3: Perhaps I missed this, but there seems to be no significant discussion of the fact that their model consistently under-predicts the observations. The quoted R-squared value is obviously being driven by the two clusters of data. This needs to be examined and discussed in detail.

OUR RESPONSE: We have added more discussion in section 3.1 on p.8.

Added: A comparison of model versus observational data and contour plots used to select relative strengths of parameter and data constraint factors are shown in Fig. 2. As can be seen, although the model generally underestimates the observation, considering the highly variable data (up to 7 orders of magnitude), the inverse model does a reasonable job of recovering the observational data ($R^2 = 0.88$).

**Reference:**

Abramson, L., Lee, C., Liu, Z., Wakeham, S., and Szlosek, J.: Exchange between suspended and sinking particles in the northwest Mediter- ranean as inferred from the organic composition of in situ pump and sediment trap samples, Limnol. Oceanogr., 55, 725–739, 2010.

Armstrong, R. A., Lee, C., Hedges, J. I., Honjo, S., and Wakeham, S. G.: A new, mechanistic model for organic carbon fluxes in the ocean based on the quantitative association of POC with ballast minerals, Deep-Sea Res. II, 49, 219–236, 2002.

Alldredge, A. L. and Gotschalk, C.: In situ settling behavior of marine snow, Limnology and Oceanography, 33, 339–351, 1988.

Asper, V. L., Honjo, S., and Orsi, T. H.: Distribution and transport of marine snow aggregates in the Panama Basin, Deep Sea Research Part A. Oceanographic Research Papers, 39, 939–952, 1992.

Clegg, S. L. and Whitfield, M.: A generalized model for the scavenging of trace metals in the open ocean-II. Thorium scavenging, Deep-Sea Res. I, 38, 91–120, 1991.

Cochran, J., Buesseler, K. O., Bacon, M. P., and Livingston, H. D.: Thorium isotopes as indicators of particle dynamics in the upper ocean: results from the JGOFS North Atlantic Bloom experiment, Deep-Sea Res. I, 40, 1569–1595, 1993.

Cochran, J. K., Buesseler, K. O., Bacon, M. P., Wang, H. W., Hirschberg, D. J., Ball, L., Andrews, J., Crossin, G., and Fleer, A.: Short-lived thorium isotopes (234Th, 228Th) as indicators of POC export and particle cycling in the Ross Sea, Southern Ocean, Deep-Sea. Res. II, 47, 3451–3490, 2000.

Fowler, S. W. and Small, L. F.: Sinking rates of euphausiid fecal pellets, Limnology and Oceanography, 17, 293–296, 1972.

Komar, P. D., Morse, A. P., Small, L. F., and Fowler, S. W.: An analysis of sinking rates of natural copepod and euphausiid fecal pellets, Limnology and Oceanography, 26, 172–180, 1981.

Marchal, O. and Lam, P. J.: What can paired measurements of Th isotope activity and particle concentration tell us about particle cycling in the ocean?, Geochim. Cosmochim. Acta, 90, 126–148, 2012.

Murnane, R. J.: Determination of thorium and particulate matter cycling parameters at station P: a reanalysis and comparison of least squares techniques, J. Geophys. Res., 99, 3393–3405, 1994.

McCave, I. N.: Vertical flux of particles in the ocean, Deep-Sea Res. I, 22, 491–502, 1975.

McDonnell, A. M. P. and Buesseler, K. O.: Variability in the average sinking velocity of marine particles, Limnol. Oceanogr., 55, 2085–2096, 2010.

Pilskaln, C. H., Lehmann, C., Paduan, J. B., and Silver, M. W.: Spatial and temporal dynamics in marine aggregate abundance, sinking rate and flux: Monterey Bay, central California, Deep Sea Research Part II: Topical Studies in Oceanography, 45, 1803–1837, 1998.

Yacobi, Y. Z., and Zohary, T., Kress, N., Hecht, A., Robarts, R. D., Waiser, M., Wood, A. M., Li W. K. W.: Chlorophyll distribution throughout the southeastern Mediterranean in relation to the physical structure of the water mass. J. Mar. Sys. 6: 179-189, 1995.

Wang, W.-L., Armstrong, R. A., Cochran, J. K., and Heilbrun, C.: 230Th and 234Th as coupled tracers of particle cycling in the ocean: A maximum likelihood approach, Deep Sea Res. I, 111, 61–70, 2016.

Wang, W.-L., Lee, C., Cochran, J. K., Primeau, F. W., and Armstrong, R. A.: A novel statistical analysis of chloropigment fluxes to constrain particle exchange and organic matter remineralization rate constants in the Mediterranean Sea, Mar. Chem., 2017.

---

## Author Comment (AC3) · 10 May 2018

Reply to editor,
May 8, 2018

Dear Editor,

We received comments from two anonymous reviewers about our BGD submission "Inferring particle dynamics in the Mediterranean Sea from in-situ pump POC and chloropigment data using Bayesian statistics (BG-2018-6)". The first reviewer was concerned that "the whole text is largely driven by methodological aspects" and questioned two model assumptions (steady state and linear aggregation). The second reviewer submitted the exact same comments as in the "Access review", which we had already replied to. We have posted our comments on line, but they were already addressed in the revised manuscript that is now posted. Perhaps he accidentally uploaded the wrong file?

Before we answer the first reviewer's key questions, we should restate the purpose of this study. First, we introduce a new Bayesian method to infer particle dynamic rate constants from new geochemical tracers (chloropigments) sampled using commonly used large volume pumps. Wang et al. (2017) suggested the use of chloropigments as new tracers to model particle dynamics, which opens a new door to the community. In that study, chloropigments were sampled using settling velocity sediment (SV) traps. However, SV traps have not been widely used due to lack of the commercial availability. In contrast, large volume pumps are widely used, but there is no method for using chloropigment data from large volume pumps. This study presents such a method. Second and more importantly, by comparing results obtained using thorium and chloropigment tracers collected using the same sampling technique, and using the same tracer but different sampling methods, we found that the tracer has a greater influence on parameter estimations than does the sampling method. Overall, either of the aspects deserves publication in BG.

Now we answer the two key questions raised by the first reviewer. 1) "the whole text is largely driven by methodological aspects", we actually just replied to this question: developing this method is what the study aimed to do. 2) "linear aggregation and steady state". The reviewer does not believe that the "linear aggregation" and "steady state" assumptions are valid. First, we argue that linear aggregation is not without merit, especially when aggregation is between the same kinds of particles. As the reviewer stated in his/her comments, "particles have to meet in order to aggregate". Therefore, aggregation has to be related in some way to the concentration of particles, no matter whether the particle concentration is high or low. By assuming a first-order reaction, we have

$$\text{aggregation} = \alpha[P],$$

where $\alpha$ is an aggregation rate constant, and $[P]$ is the particle concentration. Higher particle concentration results in a higher chance to collide, and thus more aggregation. In most empirical studies, including with the current data, we have no way to build and test a complicated model such as that in Jackson (1990). Also, unlike in Jackson (1990), we are dealing with particles below the euphotic depth, so do not consider algal division and other euphotic zone processes. Many previous models have assumed first-order reaction kinetics and show reasonable results

(Murnane et al., 1990; Clegg and Whitfield, 1991; Marchal and Lam, 2012; Lerner et al., 2016, 2017). Second, we assume "steady state" since we are using only one concentration profile. We do not have enough information to build a non-steady-state model. To achieve our second goal, to compare the influence of sampling techniques and tracers on parameter estimations, we have to make the same assumptions as we did in previous studies (Wang et al., 2016; 2017).

We realize that there will always be disagreements among researchers, and certainly particle dynamics has more than its share of "sides", but we feel that we have addressed the reviewers' major concerns and hope that you will agree.

Sincerely,

Weilei Wang